# H-GAP: Humanoid Control with a Generalist Planner

**Zhengyao Jiang** [*, 1, 5], **Yingchen Xu** [*, 1, 2], **Nolan Wagener** [3], **Yicheng Luo** [1], **Michael Janner** [4],
**Edward Grefenstette** [1], **Tim Rocktäschel** [1], **Yuandong Tian** [2]
[1] University College London, [2] AI at Meta, [3] Georgia Institute of Technology,
[4] University of California at Berkeley, [5] Weco AI

## Abstract

Humanoid control is an important research challenge offering avenues for integration into human-centric infrastructures and enabling physics-driven humanoid animations. The daunting challenges in this field stem from the difficulty of optimizing in high-dimensional action spaces and the instability introduced by the bipedal morphology of humanoids. However, the extensive collection of human motion-captured data and the derived datasets of humanoid trajectories, such as MoCapAct, paves the way to tackle these challenges. In this context, we present Humanoid Generalist Autoencoding Planner (H-GAP), a state-action trajectory generative model trained on humanoid trajectories derived from human motion-captured data, capable of adeptly handling downstream control tasks with Model Predictive Control (MPC). For 56 degrees of freedom humanoid, we empirically demonstrate that H-GAP learns to represent and generate a wide range of motor behaviours. Further, without any learning from online interactions, it can also flexibly transfer these behaviors to solve novel downstream control tasks via planning. Notably, H-GAP excels established MPC baselines that have access to the ground truth dynamics model, and is superior or comparable to offline RL methods trained for individual tasks. Finally, we do a series of empirical studies on the scaling properties of H-GAP, showing the potential for performance gains via additional data but not computing. Code and videos are available at https://ycxuyingchen.github.io/hgap/.

## 1 Introduction

Humanoid control stands as a pivotal realm in the integration into human-centric infrastructures and the generation of physics-driven animations (Peng et al., 2022). Yet, proficient humanoid control is challenging due to the inherent instability, discontinuity, and high dimensionality of the system. As a result, learning humanoid control policies from scratch not only incurs substantial computational costs but also frequently results in unintended behaviours that do not resemble human-like actions (Heess et al., 2017; Song et al., 2019).

The abundance of motion capture (MoCap) data provides a strong bedrock to address humanoid control. MoCap data demonstrates kinematic information about desirable sequences of poses assumed by the human body during various motions. By leveraging these demonstrations, it becomes possible to learn natural motions that can be used for some task of interest (Merel et al., 2017; Peng et al., 2018; 2021). However, existing methods leveraging the MoCap-derived data either need extra online interactions or result in specialized models tailored to particular tasks and reward functions (Merel et al., 2017; 2019; Hasenclever et al., 2020; Peng et al., 2022). Despite generating natural behaviours, such a strategy demands extensive online interactions, limiting its applicability in real-world, physical robots. In contrast, offline RL promises policy training without further online interactions, albeit traditionally confined to small-scale datasets and less complex robot morphologies, and offering limited transferability of learned knowledge and skills across tasks (Levine et al., 2020; Prudencio et al., 2023; Lange et al., 2012; Ernst et al., 2005).

---

* Equal contribution. Correspondence to yingchen.xu.21@ucl.ac.uk.

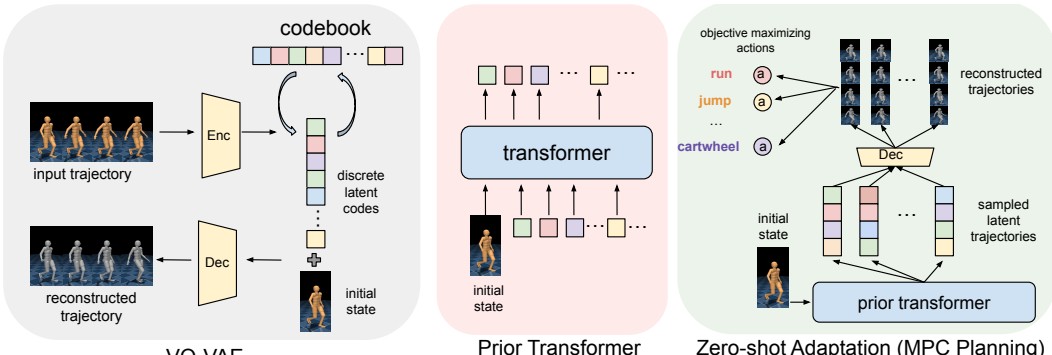

Figure 1: Overview of H-GAP. Left: A VQ-VAE that discretizes continuous state-action trajectories. Middle: A Transformer that autoregressively models the prior distribution over latent codes, conditioned on the initial state. Right: Zero-shot adapation to novel tasks via MPC planning with learned Prior Transformer, underscoring H-GAP's versatility as a generalist model.

This work aims to leverage the state-action trajectories derived from MoCap data for general-purpose humanoid control, without any online interactions. Motivated by the success of large generative models in fields such as natural language processing and computer vision, we introduce Humanoid Generalist Autoencoding Planner (H-GAP), a generalist model for humanoid control trained on the large-scale MoCapAct dataset (Wagener et al., 2022). Similar to Trajectory Autoencoding Planner (TAP) (Jiang et al., 2023c), H-GAP adeptly models the distribution of state-action sequences, conditioned on the initial state, and can be employed in downstream control tasks using Model Predictive Control (MPC). However, H-GAP differs from TAP in its focus and utility. TAP is an offline RL method specialized for specific tasks, while H-GAP is a generalist model for humanoid control. In terms of modeling, TAP captures state, action, reward, and return while H-GAP only models state and action dynamics. As a result, H-GAP places greater emphasis on accurate state prediction than TAP, which uses state prediction only as an auxiliary loss. H-GAP is designed to handle vastly larger and more varied datasets, with scales over 1,000 times greater than the single-task data used for TAP. See Fig. 1 for an overview of our framework and Appendix A for a more detailed comparison between H-GAP and TAP.

Our experiments unveil compelling insights into the performance and scalability of H-GAP. We show that H-GAP, as a generalist model, can faithfully represent a wide range of motor behaviours from the training data. When deployed for a series of downstream control tasks, we show that the same H-GAP model either outperforms or is comparable to existing offline RL methods (Wang et al., 2020; Kostrikov et al., 2022; Fujimoto & Gu, 2021; Jiang et al., 2023c) that train a specialist policy for each task, offering a more flexible and adaptable solution. Furthermore, H-GAP outperforms conventional MPC strategies, such as MPPI (Williams et al., 2017), which have access to the ground truth model (simulator). This distinction is potentially attributed to H-GAP's incorporation of a learned latent action space and a robust action prior, which are absent in traditional MPC approaches. In terms of scaling, H-GAP follows the trajectory of most generative models, where prediction accuracy and imitation performance improve proportionately with model size. Nevertheless, our analysis reveals mixed outcomes regarding the model's downstream performance as the model's size increases. This discrepancy stems from the larger model's reduced sampling diversity, thereby adversely impacting its steerability for downstream tasks. Furthermore, we discover that H-GAP benefits significantly from larger and more diverse training datasets. This observation provides an incentive for the expansion of human MoCap datasets, encompassing a broader range of real-world scenarios, thereby driving further progress in the field.

## 2 METHODOLOGY

In this section, we explain each component of H-GAP: the discretization of state-action sequences, modelling the prior distribution over latent codes, and the planning process with Model Predictive Control (MPC). We leave more low-level details and hyperparameters in Appendix B.

In general, H-GAP operates as a state-conditional trajectory generative model. Given an initial state $s_1$ and sequence length $T$, it allows for the sampling of complete state-action sequences, denoted as $\tau$, in accordance with the conditional probability $p(\tau|s_1)$:

$$\tau = (s_1, a_1, s_2, a_2, \ldots, s_T, a_T) \sim p(\tau|s_1). \tag{1}$$

For humanoid control trained on MoCapAct data, the states $s$ are egocentric proprioceptive information and actions $a$ are desired joint angles (which are converted to joint torques via pre-defined PD controllers on the humanoid). Both states and actions are continuous and high dimensional, which makes precise generative modelling rather challenging.

## 2.1 DISCRETIZING STATE-ACTION SEQUENCES

In order to model a highly diverse multimodal distribution of trajectories, discretization is a widely adopted solution (Janner et al., 2021; Reed et al., 2022; Jiang et al., 2023c; Schubert et al., 2023; Brohan et al., 2023; Jiang et al., 2023b). In particular, Jiang et al. (2023c) propose an efficient discretization based on Vector-Quantized Variational Autoencoders (VQ-VAE), which is particularly powerful for high-dimensional state-action space. We follow a similar approach and learn an encoder and decoder, illustrated on the left component in Fig. 1. Unlike TAP which models state, action, reward and return, our encoder only takes state-action sequences as input:

$$f_{\text{enc}}(s_1, a_1, s_2, a_2, \ldots, s_T, a_T) = (z_1, z_2, \ldots, z_{MT/L}), \tag{2}$$

where $L$ is the transition chunk size and $M$ is the number of latent codes assigned per chunk. Both of them will be described in detail later.

To break it down: the encoder first concatenates the input states and actions into $T$ transition vectors. It then applies a sequence model, in this case, several 1D causal convolutional layers followed by a causal Transformer decoder block, to obtain $T$ transition feature vectors. After that, it divides these transition feature vectors into $T/L$ chunks, each with $L$ transitions, introducing temporal abstraction over the original sequence. A max pooling is then applied to the vectors in the same chunk.

Applying vector quantization directly on these chunk feature vectors can lead to low reconstruction accuracy due to their high dimensionality. To remedy this, we use $M$ latent codes to describe each chunk feature vector. We split each chunk feature vector into $M$ code embeddings, resulting $MT/L$ embeddings $(x_1, \ldots, x_{MT/L})$ in total. These vectors are then mapped into $MT/L$ latent codes $(z_1, \ldots, z_{MT/L})$ by finding their nearest neighbors in a learned codebook $e \in \mathbb{R}^{K \times D}$ according to $\ell_2$ similarity, where $K$ is the size of the discrete latent space and $D$ is the dimensionality of each latent code $e_k$:

$$z_i = e_k, \text{ where } k = \arg\min_j \|x_i - e_j\|. \tag{3}$$

The decoder predicts the reconstructed trajectories based on the initial state and latent codes:

$$f_{\text{dec}}(s_1, z_1, z_2, \ldots, z_{MT/L}) = (s_1, \hat{a}_1, \hat{s}_2, \hat{a}_2, \ldots, \hat{s}_T, \hat{a}_T). \tag{4}$$

The decoding process can be seen as the inverse of the encoding process, except that the initial state $s_1$ is merged into the embeddings of the codes with a linear projection before decoding.

The autoencoder is trained by jointly minimizing the reconstruction loss, the Vector Quantisation (VQ) loss that brings the latent codes $e_i$ closer to the encoder outputs $\|\text{sg}[x] - e\|^2$, as well as the commitment loss that ensures the encoder commits to a latent code $\|x - \text{sg}[e]\|^2$, where sg stands for the "stop gradient" operator.

## 2.2 PRIOR OVER LATENT CODES

Following the discretization process, the subsequent step involves modelling the sequences of latent codes in an autoregressive manner using a decoder-only causal transformer, frequently referred to as the prior over latent codes within the context of VQ-VAE. The Prior Transformer is also conditioned on the initial state $s_1$, which is achieved by adding the state feature to all token embeddings. More specifically, the Prior Transformer models $p(z_i|z_{<i}, s_1)$, where $z_{<i} = (z_1, z_2, z_3, \ldots, z_{i-1})$ are prefix latent codes correspond to a trajectory segment starting from $s_1$. It should be noted that the sequence length $MT/L$ for modelling can be longer than the original trajectory length due to the mapping of a single transition onto multiple latent codes, thereby facilitating a more granular representation of the transitions. This is critical for H-GAP's effective planning, which will be discussed in Appendix A.

## 2.3 PLANNING WITH MODEL PREDICTIVE CONTROL

Combining the Prior Transformer and the VQ-VAE, we introduce H-GAP as a state-conditional trajectory generative model. This formation allows for the selection of optimal trajectories from those sampled from H-GAP, thereby facilitating a structured approach to planning in complex control tasks, as outlined in Algorithm 1.

The strategy for implementing H-GAP within an MPC framework is quite straightforward. Initially, we engage in top-$p$ sampling—a technique where only the top $p\%$ of the most probable samples from a given distribution are considered for selection. This is coupled with adopting a temperature parameter greater than 1. The elevated temperature ensures that the sampled trajectories exhibit a high level of diversity, facilitating a more robust planning process with a reduced sample set. However, an increased temperature can potentially generate samples with extremely low probability, affecting the modelling accuracy of H-GAP. To circumvent this issue, top-$p$ sampling is employed to filter out these low-probability, out-of-distribution samples, thereby preserving the reliability of the planning process.

In the context of MPC, an optimal trajectory is defined as the one possessing the highest objective score, calculated using a simple objective function $R(\tau) = \sum_{i=1}^{T} r(s_i)$, where $r$ represents the reward function and $\tau = (s_1, \hat{a}_1, \hat{s}_2, \hat{a}_2, \dots, \hat{s}_T, \hat{a}_T)$ denotes the sampled trajectory.

---

**Algorithm 1** H-GAP Model Predictive Control

---

**Require:** Current state $s_1$, H-GAP prior $p_\theta(z_i | z_{<i}, s_1)$ and decoder $f_{\text{dec}}$, model parameters: $T$ (steps), $M$ (codes per step), $L$ (latent steps), $N$ (number of samples), $\Upsilon$ (temperature), $\rho$ (top p threshold) , $R(\tau)$ (objective function).
1: Initialize trajectory set $\mathcal{T}_0 = \{\}$
2: **for** sample number $k = 1, \dots, N$ **do**
3:     **for** iteration $i = 1, \dots, \frac{MT}{L}$ **do**
4:         Get probability logits $o_{i,k}$ of prior $p_\theta(z_{i,k} | z_{<i,k}, s_1)$
5:         Adjust the probability with temperature $q(z_{i,k} | z_{<i,k}, s_1) \leftarrow \dfrac{e^{o_{i,k}/\Upsilon}}{\sum_{o_{j,k}} e^{o_{j,k}/\Upsilon}}$
6:         Adjust the probability with $\rho$ get $q'(z_{i,k} | z_{<i,k}, s_1)$
7:         Sample new latent codes $z_{i,k}$ according to $q'$
8:     **end for**
9:     Decode latent codes $\tau_k = f_{\text{dec}}(z_{1,k}, z_{2,k}, \dots, z_{MT/L,k}, s_1)$
10:     Append the trajectory to the set $\mathcal{T}_k = \{\tau_k\} \cup \mathcal{T}_{k-1}$
11: **end for**
12: Return the optimal trajectory $\arg\max_{\tau \in \mathcal{T}} R(\tau)$

---

## 3 EXPERIMENTS

Our experiments aim to answer the following questions: **(Q1)** How well does H-GAP represent the motor behaviours in the training data? **(Q2)** How well can H-GAP compose and sequence the learned motion priors to solve novel downstream control tasks? **(Q3)** How does H-GAP benefit from model and dataset scaling?

**Data.** We use the MoCapAct dataset (Wagener et al., 2022), which contains over 500k rollouts with a total of 67M environment transitions (corresponding to 620 hours in the simulator) from a collection of expert MoCap tracking policies for a MuJoCo-based simulated humanoid, that can faithfully track 3.5 hours of various recorded motion from CMU MoCap dataset (CMU, 2003). The dataset covers a wide range of human motor behaviours including walking, running, and more complex movements like cart-wheeling.

### 3.1 IMITATION LEARNING EVALUATION

To evaluate how well H-GAP represents the various motor priors in the training data, we design an imitation learning task where the objective is to match the humanoid poses in some reference trajectories. Similar to the multi-clip tracking task (Hasenclever et al., 2020), at the beginning of

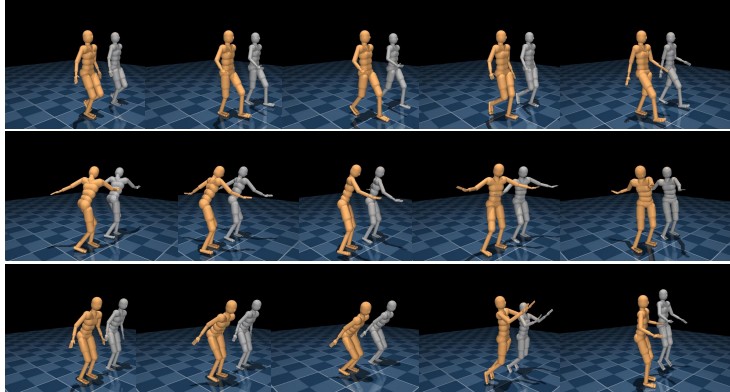

Figure 2: MoCap imitation task with simulated humanoid controlled by H-GAP (bronze) and offset reference pose (grey). Conditioned solely on an initial state, the H-GAP agent can faithfully follow the reference trajectories in a rather long horizon.

each episode, the humanoid is initialized to a pose randomly sampled from a reference trajectory. An episode is terminated when the divergence from the reference poses exceeds some predetermined threshold, or when the reference trajectory terminates. Unlike the multi-clip tracking task, we do not provide the future target reference poses to the agent since our focus is on evaluating the learned prior motor distributions solely conditioned on the initial state. To induce imitation behaviour for H-GAP, we simply greedily choose the most likely code given by the prior autoregressively and decode the action. In order to prevent the agent from succeeding just by memorising the dataset, we add a Gaussian action noise with a standard deviation of $0.01$. As shown in Figure 2, H-GAP faithfully generates a wide range of motor behaviours across multiple clips when prompted with the corresponding initial states.

In order to quantitatively evaluate performance, we train behaviour cloning (BC) (Hussein et al., 2017) agent using the entire MoCapAct dataset, which uses a 5M parameter MLP as its architecture. We compare average episodic clip tracking returns and lengths between H-GAP and BC, utilizing 12 sampled clips that showcase diverse motions from the training set. The results, as illustrated in Table 1, unequivocally demonstrate H-GAP's substantial

Table 1: Imitation Performance

|         | MLP BC       | H-GAP        |
|---------|--------------|--------------|
| Returns | 29.38 ± 2.43 | 46.02 ± 1.57 |
| Length  | 49.34 ± 3.91 | 69.11 ± 2.62 |

performance advantage over the BC baseline. Higher episodic returns indicate that the generated trajectories closely resemble the reference trajectories, whose initial states are used as prompts. In addition to tracking rewards, we also consider the episode length as an additional surrogate metric for imitation. This is because an episode concludes when the agent's deviation from the reference trajectory surpasses a predefined threshold. The results are not unexpected, as the outputs of the MLP BC agent are determined solely by the current states, while the transformer-based H-GAP incorporates elements of past trajectories into its decision-making process. This finding underscores H-GAP's superior ability to model the motor priors present in the dataset, and is in line with observations highlighted in prior research (Janner et al., 2021).

## 3.2 Downstream Control Experiments

Having verified H-GAP's capability of representing motor priors from the training data, we next investigate how well H-GAP can re-purpose the learned motor priors to solve novel downstream control tasks via planning. First, we elucidate the process of crafting downstream tasks and establishing the benchmarks for comparison, including offline RL and conventional MPC baselines. We compare the conceptual and empirical advantages of H-GAP over conventional approaches—highlighted in Table 2—as it demonstrates both high modeling accuracy and control performance with the capability for zero-shot adaptation without access to ground truth models.

Table 2: Comparison of different methods for humanoid control.

| Method | Offline Actor Critic | TAP | MPPI Baseline | H-GAP |
|---|---|---|---|---|
| Zero-shot Adaptation | No | No | Yes | Yes |
| Access to Ground Truth Model | No | No | Yes | No |
| Modeling Accuracy | / | Medium | / | High |
| Control Performance | Low | High | Low | High |

### 3.2.1 EXPERIMENT SETUP

**Construction of Downstream Tasks:** We design a suite of six control tasks: speed, forward, backward, shift left, rotate and jump. The reward functions are meticulously mapped from the agent's observational parameters, including features like egocentric velocities, egocentric angular velocities, body height, and the global z-axis vector from an egocentric perspective. For the speed task, the reward is determined as the norm of the velocities, which does not limit the direction of movement. The forward task rewards are based on the agent's forward velocity, while the backward task focuses on the backward velocity. In the shift left task, rewards are given for achieving velocity towards the left-hand side, and the rotate task allocates rewards based on y-axis angular velocity. The jump task rewards increases in body height, without imposing penalties for reductions.

The initial states for these tasks are sampled from a selection of MoCapAct trajectories, with Gaussian noise introduced to the velocity and angular velocity parameters to simulate realistic scenarios. Notably, the state dynamics mirror the conditions encountered in MoCapAct environments.

**Offline RL Baselines:** The offline RL strategies are characterized by their reliance on reward signals. Therefore, for each individual task, we apply the respective reward function to all MoCapAct trajectories, generating a reward-labelled dataset specific to that task. This is a fair set-up because the offline RL in this case can get access to all the data seen by H-GAP. On the other hand, compared with conventional offline RL settings with rich near-optimal data, this setting is challenging for out-of-distribution tasks like shift left or backward, where the majority of trajectories in the dataset are suboptimal.

**Conventional MPC Baseline:** To comprehensively evaluate the accuracy and action prior of the H-GAP model, we use MPPI (Williams et al., 2017), a state-of-the-art MPC method, as a baseline. This baseline has access to the ground truth dynamics model, thereby establishing an upper bound on modeling accuracy and offering a critical benchmark for evaluating the efficacy of H-GAP's MPC style control. In order to see how helpful the action prior in the MoCapAct dataset will be for the planning, we also tested as variation of MPPI that we denote $\text{MPPI}_{\mu,\sigma}$. In this variation, action noise during planning is parameterised according to the action mean and variance of the actions in the MoCapAct dataset.

### 3.2.2 EXPERIMENT RESULTS

Here we delve deep into the downstream performance as summarized in Table 3. We will organize the discussion around H-GAP in this subsection but the new setting and empirical results are also insightful for existing offline RL methods, which we will discuss in Appendix C.

**H-GAP versus Offline RL Methods:** H-GAP outperforms model-free offline RL methods like CRR (Wang et al., 2020), IQL (Kostrikov et al., 2022), and TD3-BC (Fujimoto & Gu, 2021) while matching the performance of the model-based offline RL algorithm, TAP. Unlike these offline RL methods, H-GAP is a *generalist* model. It does not specialize in particular tasks nor does it learn a critic for any long-term value estimation. Despite these apparent restrictions, H-GAP manages to outperform its model-free counterparts, which underscores the effectiveness of planning-based algorithms with learned discrete action spaces for humanoid locomotion tasks. This finding aligns with the previous and concurrent works on lower-dimensional continuous control, that 1) planning-based methods are highly effective Hansen et al. (2022a); Authors (2023) and 2) learning a discrete action space improves the control performance for model-free methods Tang & Agrawal (2020); Luo et al. (2023). In addition, as shown in Fig. 3d, H-GAP makes much more accurate state predictions compared to TAP. Its strong sequence modelling abilities make up for its lack of long-term credit assignments. In practical terms, the model's ability to handle multiple downstream tasks without

Table 3: H-GAP and baseline algorithms' performance on downstream tasks. Offline RL methods are trained with 5 seeds. The values after ± are standard error of the mean. For H-GAP*, discount on rewards are applied.

| Task | Offline RL Specialist | | | | MPC Generalist | | | |
| --- | --- | --- | --- | --- | --- | --- | --- | --- |
| | CRR | IQL | TD3-BC | TAP | MPPI | $\text{MPPI}_{\mu,\sigma}$ | H-GAP | H-GAP* |
| speed | 63.8 ± 4.2 | 64.9 ± 11.9 | 61.3 ± 4.4 | 65.5 ± 2.3 | 36.7 ± 1.2 | 55.1 ± 1.4 | 83.4 ± 1.9 | **97.45 ± 3.81** |
| rotate y | -7.5 ± 11.0 | 9.6 ± 10.8 | 60.2 ± 12.5 | 44.0 ± 2.3 | 23.9 ± 3.3 | 14.6 ± 3.1 | 89.5 ± 3.0 | **103.33 ± 5.61** |
| jump | 1.7 ± 0.5 | 1.2 ± 0.5 | 1.9 ± 0.6 | 3.3 ± 0.2 | 0.1 ± 0.0 | 2.0 ± 0.2 | 1.70 ± 0.1 | **3.71 ± 0.35** |
| forward | 27.6 ± 13.6 | 39.2 ± 18.0 | 35.5 ± 8.1 | 121.2 ± 10.9 | -0.9 ± 2.2 | 18.1 ± 3.6 | 51.1 ± 3.2 | 70.06 ± 4.83 |
| shift left | -4.0 ± 10.2 | 0.6 ± 10.7 | 25.7 ± 5.4 | 27.1 ± 1.0 | 4.4 ± 0.9 | -0.5 ± 3.1 | 13.2 ± 2.1 | 5.33 ± 2.74 |
| backward | -34.6 ± 12.2 | -21.2 ± 16.9 | **11.6 ± 9.9** | 1.7 ± 3.0 | 10.8 ± 1.7 | -20.6 ± 3.4 | 3.4 ± 3.1 | -2.81 ± 4.14 |
| Mean | 7.82 | 15.72 | 32.70 | 43.8 | 12.50 | 11.43 | 40.39 | **46.18** |

needing retraining is an advantageous feature. It not only saves computational resources but also provides a viable solution when it's impractical to train separate models for a large number of tasks.

**H-GAP versus Conventional MPC:** When pitted against the MPPI, a method under the MPC category, H-GAP exhibits superior performance despite MPPI having access to the ground truth model. The underwhelming performance of conventional MPC methods can be attributed to their lack of learned action space and strong action prior. The latent action prior learned from mocap data aids in constraining the action space, which historically has been challenging to optimize solely through sampling. Furthermore, H-GAP's constrained action space forces the plan to exhibit natural behaviours, potentially hinting at encapsulating long-term values. This approach prevents the model from adopting short-sighted reward maximization strategies, such as compromising balance for immediate high rewards, thus ensuring a more balanced and foresighted optimization strategy. With a naive Gaussian action prior, $\text{MPPI}_{\mu,\sigma}$ managed to perform slightly better on tasks that are well-represented in the dataset, for example: speed, forward and jump, while still inferior to H-GAP. Further, such a prior is harmful for other tasks which leads to mixed results compared with vanilla MPPI. This shows the complex action prior to humanoid control is not easy to capture with a single mod Gaussian.

### 3.3 Scaling

Scaling is key to the success of existing large generative models (Kaplan et al., 2020; Henighan et al., 2020; Hoffmann et al., 2022; Rombach et al., 2022; Podell et al., 2023) but has been little studied in humanoid control. This is likely due to the absence of comprehensive datasets like MoCapAct and general-purpose models. H-GAP fills this void, starting an early attempt to build a foundation model for humanoid control. We therefore investigate H-GAP's scaling properties to provide insights on further research in this direction.

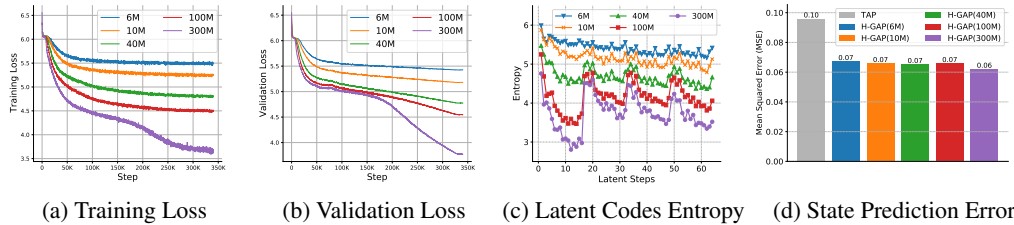

(a) Training Loss     (b) Validation Loss     (c) Latent Codes Entropy     (d) State Prediction Error

Figure 3: The graphs show training and validation losses, entropy of learned latent codes, and model prediction accuracy for different model sizes ranging from 6M to 300M parameters. Scaling up model sizes improves model validation set accuracy, which indicates that the larger models are better at modelling motor behaviours. However, the prediction accuracy improvement is marginal. The lower entropy of latent codes indicates that larger models generally produce less diverse trajectories.

**Model Scaling:** We train five different sizes of the H-GAP model, ranging from 6M to 300M parameters, utilizing 8 Nvidia V100 GPUs for each model. This scaling study draws inspiration from the work of Henighan et al. (2020), who report consistent test set performance improvement

in autoregressive transformers as the number of parameters increases across different modalities. Consistent with these findings, we observe similar trends in our validation loss, particularly noting a substantial improvement when we increase the model size from 100M to 300M parameters.

The benefits of scaling also extend to the imitation performance of H-GAP. Larger models generate more accurate control signals that successfully recover human-like motor skills. However, when we apply these scaled models to downstream control tasks, the results are mixed. Contrary to expectations, larger models do not consistently outperform their smaller counterparts. We hypothesize that this inconsistency arises from the larger models' tendency to closely replicate the trajectories present in the training dataset, which adversely affects their steerability in downstream tasks. To investigate this further, we examine the average entropy of latent codes generated by models of different sizes in a case where the initial state distribution is consistent with that of the downstream tasks. As shown in Fig. 3c, the results indicate that larger models generally produce latent codes with lower entropy, suggesting decreased diversity when generating control signals. In addition, in Fig. 3d, we compared the ability of TAP and H-GAP variations for modelling the state transitions. For an initial state from the downstream task, we sample a trajectory from the model and compare it with simulator roll-outs with the same actions in the trajectory. One can see larger models only marginally improve the prediction accuracy. Moreover, we observe degrading performance on downstream control tasks as model size scales. The intriguing results suggest that merely scaling up the model size, given the current dataset, does not offer an automatic improvement in downstream task performance even though the large model is better in (self-)supervised learning metrics.

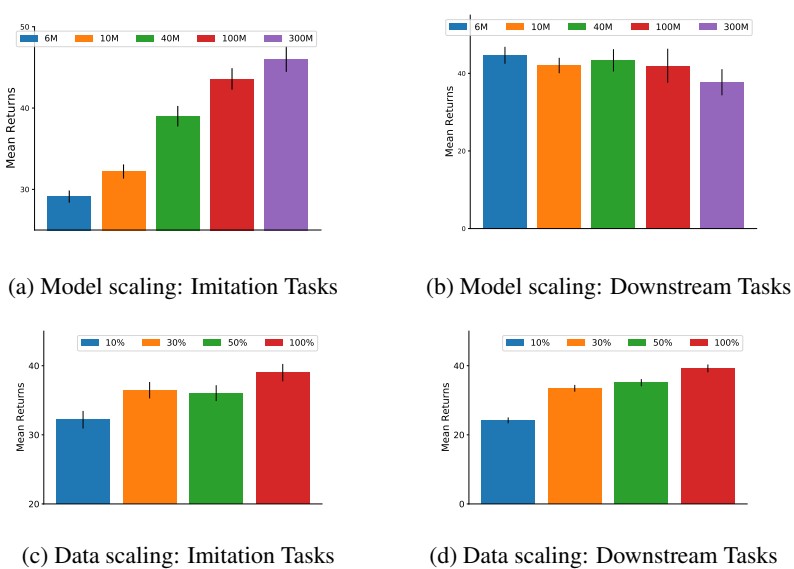

(a) Model scaling: Imitation Tasks      (b) Model scaling: Downstream Tasks

(c) Data scaling: Imitation Tasks      (d) Data scaling: Downstream Tasks

Figure 4: Scaling properties of H-GAP in terms of imitation and downstream task performances. The first row show results for different model sizes ranging from 6M to 300M parameters. Scaling up model sizes significantly improves imitation performance but surprisingly results in degrading performance in downstream control tasks. The second row show ablations for different datasets ranging from 10% to 100% of the original MoCapAct dataset. Scaling up data sizes and diversity improves model's imitation and downstream control performance.

**Data Scaling:** Given that larger models may suffer from reduced generation diversity, it stands to reason that the size and diversity of the training data could likewise influence downstream task performance. To explore this, we construct three smaller training datasets containing 10%, 30%, and 50% of the original MoCapAct dataset. These subsets range from 7M to 67M transitions. We use medium-size (40M) models for all the data scaling experiments.

As depicted in Fig. 4c and Fig. 4d, H-GAP benefits from larger and more diverse training datasets, both in terms of imitation performance and in accomplishing downstream tasks. This finding is especially compelling as it underscores the need for more comprehensive and varied human MoCap datasets to improve performance across a broader range of real-world scenarios.

## 4  RELATED WORK

**Humanoid Control** Simulated humanoids have long been a focal point in robotics research due to their remarkable physical capabilities. However, controlling these humanoids presents substantial challenges, primarily stemming from their high-dimensional action spaces and the difficulty of recovering from failure states. While attempts to learn humanoid control from scratch have historically yielded inconsistent, non-human-like motions (Heess et al., 2017; Song et al., 2019), recent research efforts have concentrated on acquiring motor primitives or reusable low-level skills derived from motion capture data (Merel et al., 2017; 2019; Hasenclever et al., 2020; Peng et al., 2022; Dou et al., 2023). Despite significant progress, the majority of these methods still necessitate online policy learning for each specific downstream task. In contrast, our proposed method showcases the capability to perform in a variety of novel downstream tasks zero-shot.

**Human Motion Generative Models** The emergence of large-scale generative models in the fields of natural language processing (Brown et al., 2020), coupled with the abundance of human motion capture data (CMU, 2003; Ionescu et al., 2014), has ignited interest in the development of human motion generative models. Many of these models require paired labeled data comprising both motion and high-level annotations such as linguistic descriptions (Jiang et al., 2023a; Ahuja & Morency, 2019; Guo et al., 2022; Tevet et al., 2023; Petrovich et al., 2021). While most prior research has centered on modeling state-only trajectories, our approach aims to learn controllable agents by also modeling action distributions.

**Model-based Reinforcement Learning** Our method closely related to model-based reinforcement learning (RL), a paradigm that involves learning a model of the environment that can be used for planning (Ebert et al., 2018; Zhang et al., 2019; Janner et al., 2019; Hafner et al., 2019; Schrittwieser et al., 2020; Hansen et al., 2022b; Chitnis et al., 2023) and/or policy learning through model-generative rollouts (Pong et al., 2018; Ha & Schmidhuber, 2018; Hafner et al., 2020; 2021; 2023). Previous works on this directly usually small Multi-Layer Perceptrons (MLPs) trained exclusively on single-task data. In contrast, our method leverages a single, large-scale transformer-based model trained on diverse datasets, which can be applied zero-shot to various downstream tasks.

**Offline Reinforcement Learning** In offline reinforcement learning (Levine et al., 2020; Prudencio et al., 2023; Lange et al., 2012; Ernst et al., 2005), the agent learns from a fixed offline dataset, aiming to achieve higher rewards than the policy that generates the data. A recent line of work treats offline RL trajectories as unstructured state-action sequences and models them with various generative models for other modalities (Janner et al., 2021; Chen et al., 2021; Jiang et al., 2023c; Janner et al., 2022; Reed et al., 2022; Jiang et al., 2023b; Schubert et al., 2023), which is most relevant to this work. The major difference between offline RL methods and H-GAP is offline RL have to be trained for specific tasks but H-GAP is trained task-unaware. H-GAP is also an early attempt to solve challenging 56 DoF humanoid control with offline data, which, as far as we know, is unexplored in offline RL and sequence modelling continuous control.

## 5  LIMITATIONS AND FUTURE WORKS

Our study identifies both the strengths and shortcomings of H-GAP, offering directions for future work. On the methodology side, while H-GAP excels in humanoid tasks, its performance may dip in sparse reward settings requiring long-term planning. Potential next steps include incorporating global information of the trajectory in the latent codes, for example, returns or goals, and investigating planning mechanisms beyond vanilla MPC, such as hierarchical planning or value learning integration. Also, H-GAP can be less effective in partially observable or stochastic environments. The model's optimistic plans, based on latent codes, may not always be realistic. Future work could explore mechanisms for more balanced planning.

With MoCapAct, we find model scaling saturation, where the validation accuracy and imitation performance still keep improving but not the downstream tasks. The experiments on different dataset sizes suggest further research could focus on data diversification and volume to forward the development of the foundation models for humanoid control.

ACKNOWLEDGEMENTS

We would like to thank Meta and the UCL CDI AWS Doctoral Scholarship for supporting this project. We also received insightful input from Siqi Liu and members of the UCL DARK Lab, among others.

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

## A   COMPARISON OF TAP AND H-GAP

Both TAP and H-GAP are trajectory generative models aimed at planning-based control, but they differ in focus and utility. This section highlights the key differences and innovations in H-GAP.

TAP is an offline RL method, where models are trained for specific tasks with their own reward functions. H-GAP, on the other hand, is a generalist model suitable for a wide range of tasks in humanoid control, offering more versatility. Moreover, H-GAP is designed to handle vastly larger and more varied datasets, with scales over 1,000 times greater than the single-task data used for TAP.

In terms of modelling and planning objectives, TAP models state, action, reward, and returns, while H-GAP only models the state and action dynamics. Accordingly, TAP's planning aims to maximize predicted rewards along with a terminal return-to-go. H-GAP focuses on maximizing an objective function based solely on rollouts. As a result, H-GAP places greater emphasis on the accuracy of state prediction than TAP. In TAP, state prediction serves mainly as an auxiliary loss. Therefore, it often maps multiple transitions to a single latent code for more efficient planning. In contrast, H-GAP assigns multiple codes to individual steps to ensure the accuracy of state reconstruction, which its objective function depends on.

H-GAP is not simply an extension of TAP within the offline RL context. It is a more streamlined, general-purpose model for generating humanoid state-action sequences. It avoids unnecessary complexities in its architecture, aiming for broad applicability across various tasks.

## B   FURTHER MODEL SPECIFICATIONS

**Mismatched Termination Conditions**   The MoCapAct dataset and the downstream tasks we consider have different termination conditions, which could result in error-prone behaviour if we were to actively predict the end of an episode. Specifically, MoCapAct trajectories terminate either due to a timeout or when the agent deviates significantly from the reference trajectories. On the other hand, in our suite of downstream tasks, termination occurs when any non-foot body parts touch the ground. To mitigate potential issues arising from these mismatched termination conditions, we opt to only sample complete trajectories during the planning phase. Importantly, the H-GAP model does not actively predict the end of the trajectory. While in theory this could trigger issues for tasks where touching the ground would result in higher rewards, this has not been observed in the suite of tasks we have constructed.

**Normalisation**   When training the VQ-VAE component with Mean Squared Error (MSE) loss, it is crucial to properly normalise the state and action dimensions to prevent large loss bumps, particularly for those dimensions with a higher range. In this work, we use the dataset mean and standard deviation for this normalisation.

**Learning Rate Annealing**   For the annealing of the learning rate, we adopt a similar pattern as proposed by (Janner et al., 2021): a linear warm-up phase followed by a cosine decay until the learning rate reaches 10% of its initial value. This scheme generally works well for smaller models. However, for the largest models with 300M parameters, some unusual training curves were observed, as shown in Fig. 3a and Fig. 3b. This phenomenon might also be attributed to the learning rate, as a smaller learning rate has been found to accelerate the learning of larger models.

**Hyperparameter**   A comprehensive list of hyperparameter settings used in our experiments can be found in Table 5.

## C   DISCUSSIONS ON BASELINES

**CRR and IQL**   Many offline RL methods have been developed in settings where the dataset contains nearly optimal trajectories, often generated by an online learning agent optimising for the same task. This makes simple behaviour cloning of high-quality trajectories in the dataset quite effective for a wide range of benchmarks (Chen et al., 2021). This has also motivated design choices that employ behaviour cloning weighted by the advantage, as calculated by a critic. However, we find

that methods relying on such weighted BC designs, such as CRR and IQL, perform poorly on the MoCapAct relabelled datasets and their corresponding downstream tasks. This underperformance is especially pronounced for out-of-distribution tasks like 'backward', 'rotate_y', and 'shift left'. This may be because, after relabelling, the majority of actions in this diverse dataset are irrelevant to the task, making simple behaviour cloning insufficient for guiding the learned policy optimally.

**TD3-BC**  Conversely, TD3-BC, which takes gradients from the critic to directly optimise the target, consistently outperforms CRR and IQL. Remarkably, even when compared with strong sequence modelling methods like TAP and H-GAP, TD3-BC holds its own, particularly on out-of-distribution tasks such as 'rotate_y', 'shift left', and 'backward'.

**TAP**  Among all the offline RL specialist methods, TAP stands out as significantly better than the others. In 'forward' tasks, TAP even manages to keep moving forward until timeout, resulting in extremely high returns. This aligns well with findings by Jiang et al. (2023c), where TAP's advantages over baselines increase as the state-action dimensionality grows. In their work, the tasks had a maximum of 24 degrees of freedom. In our study, we pushed this to 56, confirming that the benefits of a learned action space are particularly useful for high-dimensional control tasks.

**Hyperparameters**  Detailed hyperparameters for all the baselines used in our experiments can be found in Table 4.

Table 4: Hyperparameters for Baseline Models

| Model | Hyperparameter | Value |
|---|---|---|
| CRR | Discount | 0.99 |
| | Number of layers | 4 |
| | Number of units | 512 |
| | Learning rate | $1 \times 10^{-4}$ |
| | Training steps | 50k |
| IQL | Discount | 0.99 |
| | Temperature | 0.1 |
| | $\tau$ | 0.005 |
| | Expectile | 0.9 |
| | Learning rate | $1 \times 10^{-4}$ |
| | Training steps | 50k |
| TD3-BC | Discount | 0.99 |
| | $\tau$ | 0.005 |
| | BC $\alpha$ | 2.5 |
| | Learning rate | $1 \times 10^{-4}$ |
| | Training steps | 50k |
| TAP VQVAE | Batch size | 128 |
| | Training steps | 380k |
| | Modelling horizon $T$ | 24 |
| | Chunk size $L$ | 3 |
| | Code per chunk $M$ | 1 |
| | Number of codes $K$ | 512 |
| | Learning rate | $2 \times 10^{-4}$ |
| | Discount | 0.99 |
| | Reward weight | 1.0 |
| | Value weight | 1.0 |
| TAP Prior | Dimension | 512 |
| | Number of heads | 4 |
| | Number of layers | 4 |
| | Training steps | 380k |
| | Learning rate | $2 \times 10^{-4}$ |
| | Position embeddings | Absolute |

Table 5: Hyperparameters for H-GAP Model

| Component and Size | Hyperparameter | Value |
|---|---|---|
| VQ-VAE | Batch size | 128 |
| | Training steps | $10^7$ |
| | Modelling horizon $T$ | 16 |
| | Chunk size $L$ | 4 |
| | Code per chunk $M$ | 16 |
| | Number of codes $K$ | 512 |
| | Learning rate | $3 \times 10^{-4}$ |
| 6M Prior | Dimension | 384 |
| | Number of heads | 3 |
| | Number of layers | 3 |
| | Training steps | $5 \times 10^7$ |
| | Learning rate | $3 \times 10^{-4}$ |
| | Batch size | 1024 |
| | Position embeddings | Absolute |
| 10M Prior | Dimension | 512 |
| | Number of heads | 4 |
| | Number of layers | 4 |
| | Training steps | $5 \times 10^7$ |
| | Learning rate | $3 \times 10^{-4}$ |
| | Batch size | 1024 |
| | Position embeddings | Absolute |
| 40M Prior | Dimension | 768 |
| | Number of heads | 6 |
| | Number of layers | 6 |
| | Training steps | $5 \times 10^7$ |
| | Learning rate | $3 \times 10^{-4}$ |
| | Batch size | 1024 |
| | Position embeddings | Absolute |
| 100M Prior | Dimension | 1024 |
| | Number of heads | 8 |
| | Number of layers | 8 |
| | Training steps | $5 \times 10^7$ |
| | Learning rate | $3 \times 10^{-4}$ |
| | Batch size | 1024 |
| | Position embeddings | Absolute |
| 300M Prior | Dimension | 1536 |
| | Number of heads | 12 |
| | Number of layers | 12 |
| | Training steps | $10^8$ |
| | Learning rate | $3 \times 10^{-4}$ |
| | Batch size | 1024 |
| | Position embeddings | Absolute |

Table 6: Hyperparameters for MPC Planning

| Hyperparameter | Value |
|---|---|
| Sampling threshold $\rho$ | 0.99 |
| Number of samples $N$ | 256 |
| Temperature $\Upsilon$ | 4 |

# D  ABLATIONS

We use 100M parameter H-GAP models in all our ablation experiments. We report the mean and standard error over 300 random seeds for downstream control tasks and over 30 seeds for imitation tasks.

Table 7: Codes per Step.

| Codes per Step | Imitation Returns | Downstream Returns |
|:---:|:---:|:---:|
| 8 | $39.13 \pm 1.12$ | $32.84 \pm 1.29$ |
| 16 | $\mathbf{43.58 \pm 1.32}$ | $\mathbf{35.90 \pm 1.18}$ |
| 32 | $42.24 \pm 1.33$ | $32.80 \pm 0.85$ |

Table 8: Training Sequence Length and Planning Horizon

| Sequence Length | Horizon | Imitation Returns | Downstream Returns |
|:---:|:---:|:---:|:---:|
| 16 | 16 | $\mathbf{43.58 \pm 1.32}$ | $\mathbf{35.90 \pm 1.18}$ |
| 24 | 16 | $39.13 \pm 1.12$ | $32.86 \pm 1.73$ |
| 24 | 24 | $39.50 \pm 1.18$ | $30.84 \pm 1.93$ |
| 32 | 16 | $36.74 \pm 1.05$ | $28.55 \pm 1.61$ |
| 32 | 24 | $36.91 \pm 0.96$ | $24.52 \pm 1.60$ |
| 32 | 32 | $36.90 \pm 1.06$ | $21.36 \pm 1.41$ |
| 64 | 16 | $32.49 \pm 0.86$ | $23.20 \pm 1.43$ |
| 64 | 24 | $31.90 \pm 0.92$ | $22.52 \pm 1.42$ |
| 64 | 32 | $31.58 \pm 0.89$ | $21.91 \pm 1.34$ |
| 64 | 64 | $30.36 \pm 0.87$ | $19.34 \pm 1.32$ |

Table 9: Planning Hyperparameters

| Top p | Temperature | Num Samples | Returns |
|:---:|:---:|:---:|:---:|
| 0.98 | 4 | 64 | 31.87 |
| 0.98 | 4 | 128 | 33.53 |
| 0.98 | 4 | 256 | 36.39 |
| 0.98 | 8 | 64 | 29.18 |
| 0.98 | 8 | 128 | 31.87 |
| 0.98 | 8 | 256 | 38.39 |
| 0.98 | 16 | 64 | 28.89 |
| 0.98 | 16 | 128 | 29.58 |
| 0.98 | 16 | 256 | 38.17 |
| 0.99 | 4 | 64 | 32.17 |
| 0.99 | 4 | 128 | 37.29 |
| 0.99 | 4 | 256 | **38.81** |
| 0.99 | 8 | 64 | 34.52 |
| 0.99 | 8 | 128 | 31.33 |
| 0.99 | 8 | 256 | 38.06 |
| 0.99 | 16 | 64 | 28.28 |
| 0.99 | 16 | 128 | 32.20 |
| 0.99 | 16 | 256 | 34.36 |
| 0.999 | 4 | 64 | 30.44 |
| 0.999 | 4 | 128 | 30.19 |
| 0.999 | 4 | 256 | 34.54 |
| 0.999 | 8 | 64 | 27.92 |
| 0.999 | 8 | 128 | 34.14 |
| 0.999 | 8 | 256 | 33.60 |
| 0.999 | 16 | 64 | 29.40 |
| 0.999 | 16 | 128 | 29.44 |
| 0.999 | 16 | 256 | 35.11 |

