# OpenReview forum: "H-GAP: Humanoid Control with a Generalist Planner"
_ICLR.cc/2024/Conference — ICLR 2024 spotlight_

### Official Review · Reviewer_bwbh · 2023-10-23

**Soundness:** 3 good
**Presentation:** 4 excellent
**Contribution:** 3 good
**Rating:** 8
**Confidence:** 3

**Summary:**

This paper incorporates VQ-VAE, a prior transformer, and MPC planning to construct H-GAP, a state-action generative model for humanoid trajectories. Experiments show that H-GAP learns to generate a wide range of human behaviors through imitation learning and flexibly transfers to unseen downstream tasks through planning without retraining.

**Strengths:**

The proposed method outperforms MLP-backed behavior cloning in imitation learning by a large margin. Despite being a generalist policy, it also outperforms MPPI in most tasks by large margins in downstream tasks and performs reasonably well against offline RL specialist policies.

We also observe that the performance of imitation learning increases as the model and data scale up, paving the way to a possible future foundation model for humanoid control. However, we observe a drop in downstream task performance as the model size increases.

**Weaknesses:**

1. The provided link in the abstract results in 404 not found, and this submission comes without supplementary videos. This, unfortunately, makes it difficult to assess the naturalness of the generated human motion, which is critical to humanoid motion generation.
2. The paper claims that H-GAP provides a basis for a foundation model for humanoid control. However, only the performance of imitation learning improves with the model scale. The performance of downstream tasks drops for larger models. In my view, this does not constitute a foundation model's basis because supporting downstream tasks is the most important feature of a foundation model.

**Questions:**

- What are the metrics used in table 1 and table 2?
- How long does it take to train H-GAP?
- I would assume that the optimal number of samples $N$ corresponds to the length of the intended trajectory $T$. What is the choice of $N$ for your experiments when $T=M=16$ and $L=4$?
- Following up from the above question, how long does it for H-GAP to produce the reported results in table 2? How does it compare to the other methods?
- Does the temperature change throughout algorithm 1? What are the choices of $\Upsilon$ and $\rho$?

---

> ### Author Response · Authors · 2023-11-13
> **Link has been fixed**
>
> > The provided link in the abstract results in 404 not found, and this submission comes without supplementary videos. This, unfortunately, makes it difficult to assess the naturalness of the generated human motion, which is critical to humanoid motion generation.
>
> We apologize that the website didn’t work during your review. It was due to a privacy setting that was not properly set. We fixed it as soon as we noticed the issue during the reviewing period. We can confirm it is working now.
>
> This is a quick response to this particular issue and we'll address your other concerns and questions later this week (together with the response to other reviewers).

---

> ### Author Response · Authors · 2023-11-15
> **Response**
>
> > The paper claims that H-GAP provides a basis for a foundation model for humanoid control. However, only the performance of imitation learning improves with the model scale. The performance of downstream tasks drops for larger models. In my view, this does not constitute a foundation model's basis because supporting downstream tasks is the most important feature of a foundation model.
>
> Thank you for your insights. We acknowledge that the term 'foundation model' might have suggested our model is larger than it is. To clarify, we view H-GAP as an initial step towards a foundational model for humanoid control, given its generalist nature and training on a diverse dataset. We've adjusted our wording to 'an early attempt to build a foundation model for humanoid control' for clarity.
>
> The scaling properties highlighted in our paper are part of our empirical contributions, underscoring the importance of data scaling relative to model scaling. This finding suggests prioritizing data enhancement to advance the generalist approach in humanoid control. Additionally, we delve into the reasons behind the limited impact of model scaling on downstream task performance in Section 3.3, providing a deeper understanding of these dynamics.
>
> We want to highlight that our findings do not conflict with other empirical findings in foundation models for other modalities because:
> The trend of imitation learning performance and validation loss of H-GAP do align with the scaling law reported in studies on large language models (LLMs) [1, 2].
>
> Large foundation models are usually trained on a huge and diverse internet-scale dataset. For example, the whole training process of LLMs is usually only done for one or two epochs. If the training data is very limited, downstream tasks are not guaranteed to be improved just by scaling the model [3, 4].
>
>
> 	[1] Hoffmann et al. “Training Compute-Optimal Large Language Model”. 2022
> 	[2] Henighan et al. “Scaling Laws for Autoregressive Generative Modeling”. 2022
> 	[3] Muennighoff et al. “Scaling Data-Constrained Language Models”. Neurips 2023
> 	[4] Radford et. al. Robust Speech Recognition via Large-Scale Weak Supervision, 2022
>
> > What are the metrics used in table 1 (imitation performance) and table 2 (downstream performance)?
>
> The metrics used in table 1 are the average episodic clip tracking returns. The task is set up similar to the multi-clip tracking task [1], where the reward function measures the discrepancy between the agent’s joint and body positions and the target configurations in a reference trajectory. We adapt our implementation from the ReferencePosesTask from the DeepMind Control Suite [2]. We have added more explanation to the numerical results in section 3.1 of the manuscript (highlighted red).
>
> The metrics used in downstream control tasks (table 3 in the latest manuscript) are average episodic returns on downstream tasks.
>
> [1] Hasenclever et al. “CoMic: Complementary task learning & mimicry for reusable skills.” ICML 2020.
> [2] Tunyasuvunakool et al. “dm_control: Software and tasks for continuous control” 2020.
>
> > How long does it take to train H-GAP?
>
> The walk clock training time depends on the model size, number of tokens used for training and system configuration. We report the training time of the models used in most of our experiments: training a 10M parameter VQ-VAE for 6.5M tokens on 1 V100 GPU takes 9 hours. Training a 40M parameter Prior Transformer for 6.5M tokens on 8 V100 GPU on a single node takes ~20 hours. For the larger model, say, a 300M parameter Prior Transformer trained on 8 V100 GPUs for 6.5M tokens extends to approximately 45 hours.
>
> > I would assume that the optimal number of samples N corresponds to the length of the intended trajectory T. What is the choice of N for your experiments when T=N=16 and L=4?
>
> We use S=256 when reporting the results in downstream control tasks (table 3 in the latest manuscript).
>
> > Following up from the above question, how long does it take for H-GAP to produce the reported results in table 2 (downstream performance)? How does it compare to the other methods?
>
> The planning time of a 10M parameter H-GAP takes 0.29s, on a single 4090 GPU, with sample size 256. For comparison, MPPI can take about 20 seconds for a single step of decision with a trajectory ensemble size of 32, but this is probably because we lack a good parallelizable implementation. We also tried to increase the ensemble size to 64 after paper submission, but didn't observe improvement.
>
> > Does the temperature change throughout algorithm 1? What are the choices of  \upsilon and /rho
>
> The temperature is fixed throughout algorithm 1. We use \upsilon=4 and \rho=0.99 when reporting the results in downstream control tasks (table 3 in the latest manuscript). We have included a table of the planning hyperparameters and additional ablation results in the appendix.

---

> > ### Comment · Reviewer_bwbh · 2023-11-22
> >
> > The authors' reponse have addressed my concerns. Based on the revised tone and the contribution of H-GAP, I am raising my rating to 8. Good work.

---

> > > ### Author Response · Authors · 2023-11-22
> > > **Thank you for your response!**
> > >
> > > Thank you for taking the time to read our response and for acknowledging that we addressed some of your concerns!

---

### Official Review · Reviewer_kAwr · 2023-11-01

**Soundness:** 3 good
**Presentation:** 2 fair
**Contribution:** 2 fair
**Rating:** 8
**Confidence:** 5

**Summary:**

This paper tackles the challenging problem of general humanoid control, which is difficult due to high-dimensional action spaces and instability. The authors propose a Humanoid Generalist Autoencoding Planner (H-GAP) that learns a forward dynamics latent space model from the MocapACT dataset and propose model predictive control for downstream tasks. The forward dynamics model is trained using a transformer over discrete latent codes learned using a VQVAE that employs multiple codes per transition. The effectiveness of H-GAP is demonstrated through experiments on a 56 DoF humanoid across imitation benchmarks as well as analyses of its scaling properties on different model sizes.

**Strengths:**

- Trains a single prior transformer for motion completion in a more principled way to jointly learn a motion prior and forward dynamics model that can be queried autoregressively and used for MPC planning. Introduces multi-code latent space modeling with VQVAE and transformer to accurately capture complex humanoid state-action sequences.

- Analyzes scaling properties on model sizes up to 300M parameters, revealing limitations in downstream task performance despite accuracy gains.

- Demonstrates strong imitation learning on a 56 DoF humanoid, validating the model's ability to represent diverse motor behaviors from the MocapAct dataset.

**Weaknesses:**

- The paper could analyze the coverage of different motion skills by randomly sampling initial states and generating rollouts from the model. This would help quantify the diversity of motions that can be produced [1].

- For analyzing imitation abilities, the model could be conditioned on a small segment of reference trajectories rather than just initial states. This would better evaluate how well it can leverage larger context sequences. The current approach in Table 1 mainly reveals modeling/memorization capabilities rather than imitation abilities.

- Additional experiments could be designed to isolate the planning contributions of the model beyond just trajectory modeling. For example, a conditional generation task could be constructed using out-of-distribution initial states in order to see if the model is able to “catch-up” to imitate a trajectory from a different initial polse from the original trajectory.

- More complex downstream tasks requiring long-term planning could be explored to go beyond trajectory imitation. The current tasks are limited to state distributions from the training data. New tasks like going to specified locations, or following a particular heading and and orientation[1] would require guiding the motion generation process. This could reveal how well the model can piece together behaviors.

- Overall, evaluating the model on tasks and scenarios that require leveraging the full context sequence in a generative way would provide better insight into the planning abilities. The current experiments focus primarily on trajectory modeling in different flavors. (imitation based Downstream tasks and imitation benchmark)


[1] Dou, Zhiyang et al. “C·ASE: Learning Conditional Adversarial Skill Embeddings for Physics-based Characters.” *ArXiv* abs/2309.11351 (2023): n. pag.

**Questions:**

NA

---

> ### Author Response · Authors · 2023-11-15
> **Response**
>
> Thank you for your valuable feedback. We'd like to clarify a potential misunderstanding regarding our experiments. The setups for our imitation experiments (Section 3.1 and Table 1) and downstream control tasks (Section 3.2 and Table 3) are distinctly different.
>
> **Initial State Configurations**: In the imitation tasks, the initial states are derived from the trajectories of the reference motions, varying across subtasks. For downstream tasks, however, a uniform initial state distribution is used across all tasks. This setup demonstrates that H-GAP's performance exceeds mere imitation, as evidenced by non-zero sum returns in forward and backward tasks.
>
> **Action Selection Approaches**: During imitation learning, we decode latent codes greedily, without employing any planning. Conversely, in downstream tasks, we utilize Model Predictive Control (MPC) to optimize the agent's behavior for achieving the highest task rewards, which further differentiates the two experimental setups.
>
> **Further experiments**: To further address your concern on this, we run an actual imitation H-GAP agent on downstream control tasks, and the average score across tasks are 22.04, which is significantly lower than H-GAP with MPC planning (46.18).
>
> > The paper could analyze the coverage of different motion skills by randomly sampling initial states and generating rollouts from the model. This would help quantify the diversity of motions that can be produced [1].
>
> In response to your suggestion, we would appreciate further details regarding the specific distribution you have in mind for sampling.
> Uniform or Gaussian sampling in a high-dimensional state space often leads to unnatural poses and ineffective rollouts.
> On the other hand, if you're referring to sampling from the MocapAct dataset, our imitation learning experiments serve as an indirect assessment of the model's ability to capture a broad spectrum of behaviors. Here, initial states are drawn from a variety of clips, and Gaussian action noise is added for enhanced variability. If your suggestion falls outside these approaches, we'd appreciate further clarification to better address your point.
>
> > For analyzing imitation abilities, the model could be conditioned on a small segment of reference trajectories ... Table 1 mainly reveals modeling/memorization capabilities rather than imitation abilities.
>
> In addressing your suggestion for conditioning the model on reference trajectories, we interpret this in two ways: conditioning on either a longer historical context or a target future trajectory. Using a longer historical context could enhance imitation performance, but may reduce the steerability for downstream control tasks. Our focus with H-GAP is on generalist control, where imitation learning primarily serves to validate model accuracy.
>
> Conditioning on a target future trajectory, while potentially improving imitation, diverges from the norm in imitation learning, where policies typically access only current observations, not future trajectories. We believe our current approach maintains a fair balance in this context.
>
> Additionally, to counteract mere memorization by the agent, we introduce Gaussian noise into the action selection process, ensuring that the model's performance extends beyond simple replication of data.
>
> > Additional experiments could be designed to isolate the planning contributions of the model beyond just trajectory modeling. ... is able to “catch-up” to imitate a trajectory from a different initial pose from the original trajectory.
> More complex downstream tasks ... This could reveal how well the model can piece together behaviors.
> Overall, evaluating the model on... provide better insight into the planning abilities. The current experiments focus primarily on trajectory modeling in different flavors. (imitation based Downstream tasks and imitation benchmark)
>
>
> Thank you for your valuable suggestions. It appears that your concerns focus on whether H-GAP, coupled with Model Predictive Control (MPC), is only used for to imitation. To address this, we'd like to emphasize the diverse and perturbed nature of initial states used in our experiments, as detailed in Section 3.2.1. These states, often not directly aligned with the specific downstream task, necessitate adaptive behaviors from the agent. For instance, in the 'rotate y' task in this rollout video: , the agent starts from a forward-walking initial state and must adjust to execute the rotation, demonstrating the model's capability to “piece together behaviors" in a non-trivial manner.
>
> Looking ahead, we are committed to exploring long-horizon, sparse-reward tasks to further probe our model's potential as you suggested. Our approach's strength in encoding compact, high-dimensional humanoid trajectories positions us well to tackle these more complex, long-horizon challenges, demonstrating the adaptability and broad applicability of our method.

---

> > ### Comment · Reviewer_kAwr · 2023-11-23
> >
> > I thank the authors for the detailed response. This was very helpful. The Authors commitment to exploring the potential of the direction is very welcome. I have updated my score.
> >
> > Some additional comments would be to include (1) explanations on the performance drop see in larger planning horizons, (2) the compute/latency tradeoff incurred in different choices of MPC parameters (3) some standard metrics like MPJPE [1] on the current imitation tasks as well as some noisy tasks such as video tracking data. This noisy task with partial matching goal further explores the flexibility/limitations afforded by the MPC framework.
> >
> > [1] Luo, Zhengyi et al. “Perpetual Humanoid Control for Real-time Simulated Avatars.” ArXiv abs/2305.06456 (2023): n. pag.

---

> ### Author Response · Authors · 2023-11-16
> **New Reference Inclusion**
>
> We would like to also thank you for your suggestion regarding additional references on humanoid control. We have incorporated the following reference into the revised version of our manuscript:
>
> >Dou, Zhiyang et al. “C·ASE: Learning Conditional Adversarial Skill Embeddings for Physics-based Characters.” ArXiv abs/2309.11351 (2023): n. pag.

---

### Official Review · Reviewer_oGKQ · 2023-11-02

**Soundness:** 3 good
**Presentation:** 2 fair
**Contribution:** 3 good
**Rating:** 6
**Confidence:** 3

**Summary:**

This paper presents a Humanoid Generalist Autoencoding Planner (H-GAP), consisting of a VQ-VAE that maps state-action trajectories into potentially discrete codes based on MoCapAct, a transformer capable of modeling future distributions based on past observations, and an MPC planning algorithm based on multiple trajectories. Extensive experiments show that H-GAP can benefit from larger and more diverse datasets, demonstrating the potential of this line of work.

**Strengths:**

The proposed H-GAP is even simpler than the existing offline RL methods while it is general for different downstream control tasks, meaning that the algorithm does not need access to the simulator and train different high-level policies.

Experimental results also show that the proposed H-GAP can outperform offline RL policy and other traditional MPC algorithms.

**Weaknesses:**

Looking at the visualization on the website, I'm wondering why there is severe jitter compared to the ground truth sequence. I'm not sure if this is a result of the transformer not modeling temporal smoothness well, or if VQ-VAE doesn't represent state-motion clips well.

I'm still not fully convinced of the ability of the proposed method to model a variety of humanoid motions (e.g., backflips) that have more complex patterns than locomotion. While the paper emphasizes data and model scaling experiments, which I appreciate, train data is a random sampling of the dataset, not of how performance changes when more action categories are added.

I guess a potential limitation is the need to generate multiple trajectories. I would like to know how efficient the proposed method is, what the number of trajectories that typically need to be sampled, and whether these hyperparameters need to be changed to accept more trajectories as the motion database becomes more complex.

**Questions:**

I'm not sure I understand correctly that the whole framework doesn't use a simulator at all. So what's the point of modeling action trajectories here, why not just state trajectories? And if so, why don't we just follow the existing kinematics-based human motion generation?

Overall, the author's response to the concerns is needed to make the final decision. I am happy to increase the rating if my concerns are addressed.

---

> ### Author Response · Authors · 2023-11-15
> **Response**
>
> Thank you for your constructive feedback. We're pleased that you recognize H-GAP's key features: its independence from a simulator during training and its general-purpose design.
>
> To clarify, our primary focus is on continuous robotic control, not human motion animation generation. In the MPC planning process, we use predicted state rollouts to determine optimal actions, rather than to create motion animations. The videos on our website showcase this control policy in action within a simulator.
>
> While it's possible to use our model's trajectories for animation purposes, we view this as a separate and intriguing line of research. In a control context, these trajectories might appear natural but may not adhere closely enough to physical laws for effective action selection.
>
>
> > Looking at the visualization on the website, I'm wondering why there is severe jitter compared to the ground truth sequence. I'm not sure if this is a result of the transformer not modeling temporal smoothness well, or if VQ-VAE doesn't represent state-motion clips well.
>
> Your observation about the jitter in the visualizations is noteworthy and something we've also encountered in our research. These visualizations are simulator rollouts, not direct samples from the generative model. The jitter likely arises not from a lack of temporal smoothness in the sequence modeling but from control signal noise. We introduced action noise to enhance the challenge of the imitation task, preventing the agent from merely memorizing data. Additionally, the original Mocap data used in our model has inherent jitter, particularly in hand movements, due to tracking instabilities.
>
> > I'm still not fully convinced of the ability of the proposed method to model a variety of humanoid motions (e.g., backflips) that have more complex patterns than locomotion. While the paper emphasizes data and model scaling experiments, which I appreciate, train data is a random sampling of the dataset, not of how performance changes when more action categories are added.
>
> Regarding your concerns about our method's capability to model complex humanoid motions, we hope the updated video clips on our website will provide some reassurance.
> We also want to clarify that the smaller datasets used in the data scaling experiments in Section 3.3 are constructed by sampling on the clip level, not on the transition or trajectory level. Each clip in the mocapact correspond to a certain type of behaviour so we believe the experiment does show the performance improves along with qualitatively more diversed actions are added.
>
> > I guess a potential limitation is the need to generate multiple trajectories. I would like to know how efficient the proposed method is, what the number of trajectories that typically need to be sampled, and whether these hyperparameters need to be changed to accept more trajectories as the motion database becomes more complex.
>
> For all downstream control tasks and data scaling experiments, we sample S=256 trajectories. The planning process with a 10M parameter H-GAP model takes approximately 0.29 seconds on a single 4090 GPU. We acknowledge the potential for increased efficiency and are considering techniques like beam search to enhance the planning process in future developments.
>
> > I'm not sure I understand correctly that the whole framework doesn't use a simulator at all. So what's the point of modeling action trajectories here, why not just state trajectories? And if so, why don't we just follow the existing kinematics-based human motion generation?
>
> H-GAP is designed as a generative model for humanoid control, necessitating the modeling of actions, not just state trajectories, for effective control in both simulated and real-world environments. Unlike kinematics-based human motion generation methods, which lack output in executable actions, H-GAP fulfills the requirement for control tasks.

---

### Author Response · Authors · 2023-11-15
**General Response**

We appreciate the constructive feedback from all reviewers. Their comments highlight key strengths of our work:

- Reviewer oGKQ recognized H-GAP's simplicity, simulator independence, and adaptability to various tasks, noting its superiority in experimental results.
- Reviewer kAwr commended the use of a single prior transformer in H-GAP for effective motion completion and its strong imitation learning performance.
- Reviewer bwbh noted H-GAP's impressive imitation learning results, its ability to surpass existing methods in downstream tasks, and its potential as a foundational model for humanoid control.

We have addressed the questions and provided clarifications on the critiques in our individual responses to each reviewer.

In our latest draft, we've included new results demonstrating H-GAP's enhanced performance through the integration of discount factors on rewards. This update shows H-GAP outperforming all offline RL specialists in mean scores, now reflected in Table 3 as the new column 'H-GAP*'.

Additionally, we've updated the supplementary videos on our Anonymous website to showcase more qualitative results of H-GAP's capabilities.

All modifications in the paper are clearly marked in red for easy identification.

---

### Author Response · Authors · 2023-11-20
**Looking forward to your responses**

Dear reviewers,
We have submitted our rebuttal in response to the feedback provided for our paper, and we eagerly await your further comments and suggestions. Your insights are invaluable to us, and we are keen to engage in further discussions to enhance our work. We appreciate your time and expertise in this process and look forward to your responses.

---

### Meta-Review · Area_Chair_GzJ6 · 2024-01-15

**Metareview:**

This submission proposes a simple yet powerful framework for a generative model of humanoid trajectories, learned using a collection of motion-captured data. The method is straightforward: it involves training a transformer-based generator (causal transformer) combined with a sequence discretizer (VQ-VAE), followed by employing Model Predictive Control (MPC) planning with samples from the generator. By combining these modules, the model achieves state-of-the-art performance in humanoid trajectory generation, even demonstrating success in zero-shot adaptation scenarios. All reviewers provided positive feedback regarding its simplicity, presentation clarity, and results. The AC also concurs with the reviewers after a review of the draft. Consequently, the draft is recommended for acceptance.

**Justification For Why Not Higher Score:**

While the performance achievement is significant, the method is rather a combination of existing methods, without much new insight. I still believe the engineering mechanism is valuable to share with the community, and also the method can potentially generalize to other domains.

**Justification For Why Not Lower Score:**

n/a

---

### Decision · Program_Chairs · 2024-01-16

Accept (spotlight)